# Systemic Inflammatory Response Index (SIRI) at Baseline Predicts Clinical Response for a Subset of Treatment-Resistant Bipolar Depressed Patients

**DOI:** 10.3390/jpm13091408

**Published:** 2023-09-20

**Authors:** Stephen Murata, Nausheen Baig, Kyle Decker, Angelos Halaris

**Affiliations:** 1Pine Rest Christian Mental Health Services, Michigan State University, 300 68th Street SE, Grand Rapids, MI 49548, USA; 2Department of Psychiatry and Behavioral Neurosciences, Loyola University Chicago, Stritch School of Medicine, Loyola University Medical Center, Maywood, IL 60153, USA; nbaig5@luc.edu (N.B.); kdecker3@luc.edu (K.D.); ahalaris@luc.edu (A.H.); 3Stritch School of Medicine, Loyola University, Maywood, IL 60153, USA

**Keywords:** systemic inflammatory response index, inflammation, bipolar depression, treatment resistance, celecoxib, escitalopram, age, neuroprogression, mood disorders, biomarkers

## Abstract

**Background**: in a recent double-blind, placebo controlled RCT we demonstrated that selective inhibition of cyclo-oxygenase 2 (COX2) is an effective adjunctive strategy in treatment-resistant bipolar depression (TRBDD). To better clarify the mechanisms underlying TRBDD and treatment response, we conducted a retrospective exploratory analysis of the systemic inflammatory response index (SIRI = absolute neutrophils × absolute monocytes/absolute lymphocytes) in relation to other biomarkers and clinical outcomes after escitalopram (ESC), combined with the COX-2 inhibitor, celecoxib (CBX), versus placebo. **Methods**: Baseline measures of SIRI were compared between TRBDD and healthy controls (HC), and correlated with blood-based inflammatory cytokines, kynurenines, and growth factors. Post-treatment Hamilton Depression Rating Scale 17 (HAMD-17) total scores (clinical outcome) were modelled according to SIRI adjusting for demographics (including relevant interactions with SIRI), baseline depression, treatment arm, and treatment timepoint using multiple linear regression and robust linear mixed effects models. **Results**: Baseline SIRI did not distinguish TRBDD from HC groups. Baseline SIRI was significantly correlated with lower baseline MCP-1. The relationship between SIRI and HAMD-17 was significant at treatment week 8, in contrast to baseline. Finally, baseline SIRI predicted elevated post-treatment HAMD-17 scores, amongst patients with elevated depression scores at baseline. **Significance**: High pre-treatment SIRI may predict poorer depressive outcomes amongst TRBDD patients with baseline elevated depression.

## 1. Introduction

Considering the emerging link between psychiatric disorders and immune–metabolic dysregulation [1], there is a need to understand the biological mechanisms underlying various forms of depressive illness. A useful construct here is *neuroprogression*, which refers to the neurobiological correlates involved in the relapsing, remitting course of bipolar illness [1,2,3]. Immune–metabolic dysregulation is increasingly implicated in bipolar neuroprogression [1,2]. While bipolar patients exhibit elevated inflammatory burden compared to healthy controls, they also exhibit fluctuations in inflammatory and metabolic markers depending on the clinical phase of bipolar illness [4,5,6,7].

Due to its role in depressive neuroprogression generally, immune-metabolic dysregulation is also increasingly implicated in a subset of treatment-refractory forms of depressive illness [1,8,9]. In that vein, several randomized control trials (RCTs) have established preliminary support for adjunctive anti-inflammatory treatment in treatment-resistant depression [10,11,12,13,14,15,16,17]. In a recent double-blind, placebo-controlled RCT in treatment-resistant bipolar depression (TRBDD), we demonstrated that clinical remission can be induced with adjunctive immune modulatory treatment with the selective cyclo-oxygenase 2 (COX2) inhibitor, celecoxib (CBX), in conjunction with an adequately dosed selective serotonin reuptake inhibitor (SSRI), escitalopram (ESC) [18]. Since then, we have conducted several follow-up biomarker studies to clarify group differences in TRBDD compared to healthy controls [19,20,21,22,23]. Specifically, we found evidence of elevated pre-treatment levels of baseline pro-inflammatory markers including interleukin-1 beta (IL-1β) and C-reactive protein (CRP) in TRBDD compared to healthy controls [19,20]. From a metabolic standpoint, TRBDD subjects also showed abnormalities in the kynurenine pathway (KP), which diverts tryptophan (TRP) away from serotonin (5-HT) synthesis and towards various neuroactive metabolites in a cytokine-dependent manner [21]. We also explored markers involved in treatment responsiveness to adjunctive CBX in our TRBDD sample, including polymorphisms in the CRP gene and monocyte chemoattractant protein (MCP1) [22,23]. Taken together, these findings support the generally accepted concept that immune–metabolic dysfunction may contribute to the underlying biosignature of TRBDD thus indicating a likelihood of beneficial clinical response to adjunctive CBX.

The current ancillary study was conceived based on the emerging interest in differentiated circulating immune cells (derived from the complete blood count or CBC) as an economic, readily accessible index of the peripheral inflammatory response. While there are several indices derived from ratios of immune cells that are beyond the scope of this report [24,25,26], the focus of the current report is the Systemic Inflammatory Response Index (SIRI) calculated by *absolute neutrophils* × *absolute monocytes/absolute lymphocytes* [26,27]. 

Conceptually, the SIRI formula expresses the proportion of cells involved in the “innate” immune response (e.g., neutrophils, monocytes, and platelets) relative to those involved more directly in the “adaptive” or “cell-mediated” immune response (lymphocytes) [28,29,30]. Innate immunity refers collectively to mechanisms involved in the rapid, but non-specific, response of a host to a pathogen (such as chronic low-grade inflammation) [31,32], whereas adaptive or “cell-mediated” immunity refers to more targeted mechanisms tuned to the specific pathogen (such as during a viral infection or even autoimmune illness) [31].

Such indices of systemic inflammation, including SIRI, have already shown promise in other disease processes (e.g., neoplastic, cardiovascular, neurological, and infectious illness) where prognoses are difficult to make by clinical assessment alone [24,26,27,33,34,35]. In the context of psychiatry, there have been preliminary reports of abnormal counts of circulating immune cells in depressed patients with COVID-19 [36], major depressive disorder (MDD) [3], and bipolar disorder (BD) [37,38,39] where profiles of circulating immune cells may differ according to illness phase, specifically bipolar mania [40,41,42,43,44,45]. To date, no studies have explored the role of peripheral immune cells in bipolar depression, including TRBDD.

The broad objective of this ancillary biomarker study is to explore the clinical and biological relevance of SIRI in the context of TRBDD. We present three hypotheses:(1)First, based on prior reports that circulating immune cell profiles discriminate bipolar subjects in the general population [37,38,39], we hypothesize that elevated SIRI will distinguish TRBDD from healthy controls.(2)Secondly, based on the notion that SIRI is an index of peripheral inflammation, we hypothesize that baseline SIRI is associated with classical pro-inflammatory cytokines. For exploratory purposes, we also assessed relationships between SIRI and biologically pertinent neurotrophic factors and kynurenine pathway metabolites.(3)Thirdly, based on reports of within-patient changes in inflammatory profiles in bipolar patients between the depressed and euthymic and/or (hypo)manic phases [40,41,42,43,44,45], it is plausible that SIRI may also be associated with the clinical course of depression over treatment timepoints in TRBDD. We therefore hypothesized that changes in the relationship between SIRI and depressive severity (HAMD-17) will differ significantly between treatment timepoints (e.g., baseline and week 8).(4)Fourth, we hypothesized that pre-treatment (baseline) SIRI levels are associated with post-treatment clinical outcomes (depressive severity). This last hypothesis is based on the broader notion that treatment refractoriness is based, at least in part, on pre-existing immune–metabolic abnormalities in the literature and also our sample [22,23].

## 2. Materials and Methods

### 2.1. Study Population

Males and females aged between 21 and 65 who had a diagnosis of BD I or II based on the Diagnostic and Statistical Manual of Mental Disorders IV (DSM-IV) and met criteria for TRBDD with a minimum score of 18 on the 17-item Hamilton Depression Scale (HAMD-17) were considered for the study. The sample (N = 69) was 65.2% female and 65.2% white, with a mean age of 42 years (SD = 12.7). Viable participants had no other medical diagnosis. A comorbid psychiatric diagnosis, except for anxiety disorder, was an exclusionary criterion. To be classified as treatment-resistant, participants had to have previously failed at least two adequate trials of antidepressants or combination/augmentation with a mood stabilizer or atypical antipsychotic medication, as outlined by the Maudsley Staging Method (MSM) [46,47]. In our study, 80% of the patients scored between 5 and 8 (on the MSM), indicating moderate to severe treatment resistance. Patients had to be clinically stable on either a mood stabilizer and/or antipsychotic medication before entering the study. 

A history of substance use or dependence within 12 months preceding the screening visit was exclusionary. Patients were excluded in the presence of any abnormal routine laboratory examinations, a pain condition including fibromyalgia, history of peptic ulcer, uncontrolled hypertension, anemia, liver disease, kidney disease, arthritis, recurrent migraines, epilepsy, stroke, gum disease, autoimmune disease, pregnant or lactating females, and females taking oral contraceptives. Concurrent use of stimulants, anticoagulant agents, nicotine containing substances, corticosteroids, or lithium was exclusionary. Celecoxib, a COX-2 inhibitor and a widely used anti-inflammatory, has the potential to increase lithium blood levels leading to toxicity. Routine blood test results were reviewed to ensure normal ranges in complete blood count, complete metabolic panel, lipid profile, and thyroid function. Urinalysis and urine drug screen were conducted to further exclude participants with underlying infection and/or drug use. Known allergies or hypersensitivities to the study medications and concomitant pharmacologic contraindications were additional exclusion criteria. During the initial screening visit, the study protocol was presented in detail to potential participants and written informed consent was obtained as approved by the Institutional Review Board (IRB) of Loyola University Medical Center (LUMC). 

### 2.2. Healthy Controls

The original RCT design for evaluating primary clinical outcomes did not include the healthy control (HC) group [18]. For supplemental molecular analysis, an HC group was utilized from our database (included in supplemental information) to compare against TRBDD subjects. Recruitment was conducted via flyers on the LUMC campus. Volunteers were required to provide written informed consent approved by the IRB before the screening process. Screening and exclusion criteria were similar to those of the TRBDD group with the key difference being a negative history or current mental illness. Subjects were excluded if they had any current medical conditions or a significant history thereof. Regarding mental illness, HCs were excluded if there was any personal or family history in first-degree relatives for substance use and/or mental illness. HAMD-17 scores were required to be less than 5 on the evaluation. Blood samples were only required once at the initial screening, as HCs did not receive any intervention. Based on our experience, measured values are stable barring any intercurrent illness. HC subjects were enrolled if their routine laboratory tests fell within normal range. 

### 2.3. Study Design 

The study was approved by the IRB and was conducted according to the principles of the Declaration of Helsinki. Full details of the study design can be found in our primary study [18]. For ease of reference, the necessary details are summarized here as well. This was a 10-week, randomized, double-blind, placebo-controlled, two-arm study of TRBDD patients using escitalopram (ESC), in combination with an anti-inflammatory medication, celecoxib (CBX). It included an initial screening visit, a 2-week minimum washout phase, a 1-week placebo run-in phase, and an 8-week flexible dosing phase. Males and females aged between 21 and 65 who were diagnosed with TRBDD while being mentally and physically capable of consenting to the study were considered. The study was powered for 70 patients (35 in each treatment arm) to complete 8 weeks of active medication with an anticipated 10% dropout rate based on experience with our patient population in the preceding five years. One treatment arm consisted of ESC in combination with CBX (*n* = 26) while the other arm received ESC with PBO (*n* = 21). 

Screening visit 1 consisted of a physical exam, blood draws, and urinalysis to obtain CBC, CMP, thyroid function, lipid profile, hCG pregnancy test, and toxicology screen. Subjects were diagnosed with TRBDD through structured interviews using the Mini International Neuropsychiatric Interview (MINI) and the Maudsley Staging Scales. Depression severity and associated symptoms were quantified using the 17-item Hamilton Rating Scale for Depression (HAMD-17) [48], Hamilton Anxiety for Anxiety (HAMA) [49], Clinical Global Impressions (CGI) [50], and Columbia Suicide Severity rating Scale (CSSRS) [51]. Psychiatric and family histories were obtained through interviewing and with focused questionnaires. After the 1-week placebo run-in phase, subjects were evaluated to effectively rule out placebo responders. Successful placebo non-responders were randomized in a 1:1 fixed assignment ratio to receive either ESC + CBX or ESC + PBO. The approach to stratification included two age groups, (21–45) and (46–65), plus binary genders (male, female), and group assignment was based on a pharmacy-generated randomization code. The randomization code was generated by the institutional pharmacist. Study medications were prepared by the pharmacist, sealed in envelopes, administered to subjects, and returned to the study coordinator at each study visit to ensure compliance. 

CBX was dosed at 200 mg twice daily, while ESC was started at 10 mg per day and later titrated up to 10 mg twice per day. However, several exceptions became necessary to optimize clinical response and minimize adverse side effects: in the ESC + CBX arm, 6 patients were dosed at 10 mg of ESC and 1 patient at 30 mg ESC; in the ESC + PBO arm, 3 patients were dosed at 10 mg ESC, and 2 patients were dosed at 30 mg and 40 mg ESC. Along with the study medication, patients were prescribed one or more of the following medications for mood stabilization as indicated: quetiapine, lamotrigine, divalproex sodium, buspirone, topiramate, ziprasidone, oxcarbazepine, gabapentin, carbamazepine, asenapine, risperidone, olanzapine, aripiprazole, zolpidem, and lurasidone. 

A minimum score of 18 on the HAMD-17 scale was required for enrollment. “Responders” to treatment were defined as those whose baseline HAMD-17 scored dropped by at least 50% by week 8 but was still above a score of 7. “Non-responders” were defined as subjects whose HAMD-17 scores dropped less than 50% by week 8. Remission was defined as a score of ≤7 on HAMD-17 at treatment endpoint (week 8). 

### 2.4. Laboratory Measurements of Biomarkers and Calculation of SIRI

CBC with differential was drawn at baseline and week 8 between 9 and 10 AM to control diurnal variations. SIRI was calculated as follows: SIRI = (absolute neutrophil count × absolute monocyte count)/absolute lymphocyte count. 

The following groups of blood biomarkers were included in the correlational analyses. All of these biomarkers had been measured during the course of the original clinical study: inflammation biomarkers including C-reactive protein (CRP) and the interleukins IL-1A, IL-2, IL-4, IL-6, IL-8, IL-10, and IL-18, interferon gamma (IFN-γ), tumor necrosis factor alpha (TNF-α), and the chemokine called monocyte chemoattractant protein 1 (MCP-1); the neurotrophins epidermal growth factor and vascular endothelial growth factor; and the entire kynurenine pathway including tryptophan (TRP), kynurenine (KYN), kynurenic acid (KYNA), 3-hydroxykynurenine (3-HK), anthranilic acid (AA), xanthurenic acid (XAN), picolinic acid (PIC), quinolinic acid (QUIN), and ratios thereof as previously reported in the literature [21,52]. These biomarkers were determined at baseline and end of treatment (week 8). 

Plasma samples were analyzed using Zymutest High Sensitivity CRP enzyme-linked immunosorbent assay (ELISA) kit (Hyphen Biomed^®^, Neuville-sur-Oise, France). This is a highly sensitive “one step” sandwich ELISA technique specific for human CRP. Levels of cytokines and growth factors were measured using a Randox Cytokine and Growth Factors High-Sensitivity Array assay (Randox^®^, London, UK). This is a chemiluminescent immunoassay that operates on a sandwich principle similar to that used in ELISA. Procedures were followed according to the protocols for both assays. Kynurenine pathway metabolites were measured by Ultra Performance Liquid Chromatography/Mass Spectrometry (UPLC-MS), using a Waters Acquity UPLC connected to a Xevo TQ MS triple-quadrupole mass spectrometer, equipped with a Z-spray ESI ion source (Waters, Milford, MA, USA). Separation was carried out using a Kinetex XBC18, 2.6 μm, 2.1 × 150 mm column (Phenomenex, Torrance, CA, USA).

### 2.5. Clinical Outcome Variable

Depression severity was measured using the HAMD-17 rating scale administered at baseline and treatment endpoint (week 8). In the primary clinical study (Halaris et al., 2020) [18], the authors utilized clinical outcome variables defined categorically (dichotomous), including treatment “response” defined as ≥50% reduction of HAMD-17 total score between baseline and week 8, and treatment “remission” defined as HAMD-17 total score ≤7 at treatment endpoint (week 8) regardless of baseline HAMD-17. However, based on the foreseen limitations of a secondary retrospective biomarker analysis (including limited sample size), the clinical outcome utilized in this report was depressive severity (HAMD-17 total score) measured continuously, including at either timepoint (baseline or week 8).

### 2.6. Statistical Analysis 

Statistical analysis was conducted using R-3.6.3. Continuous clinical variables and biomarkers were transformed using natural log transformation to meet the assumptions of normality. Variables that did not require log transform included age and HAMD-17 total scores at baseline or week 8.

To test hypothesis 1, we conducted a group comparison of all baseline sample characteristics according to clinical subgroup, which was a categorical (nominal) variable with 3 categories: healthy controls (HC), TRBDD subjects in the treatment arm (ESC + CBX), and TRBDD subjects in the placebo arm (ESC + PBO). For continuous variables we used Kruskal–Wallis rank sum test and post-hoc pairwise comparisons when group differences were significant (notated with superscripts). Chi-square was used and Fisher’s exact test was used for categorical variables. We used alpha = 0.05 as the significance threshold for all tests. Post-hoc Benjamini–Hochberg (BH) analyses were applied to adjust for false discovery rate (FDR). 

To test hypothesis 2, we utilized Spearman’s correlations (rho) to test univariate relationships between SIRI (baseline) and all sample characteristics (demographic variables, HAMD-17 at baseline and week 8 timepoints, and pertinent biomarkers at baseline). Spearman’s was used rather than Pearson’s because most continuous variables were log transformed. Post-hoc Benjamini–Hochberg analyses were applied to adjust for false discovery rate (FDR).

To test hypothesis 3, we utilized robust linear mixed-effects modelling to describe HAMD-17 total score (outcome) according to SIRI—to—timepoint interaction (fixed effect) and timepoint (random effect), adjusted for sex, age, BMI and treatment arm. Treatment timepoint, sex, and treatment arm were dichotomous dummy coded variables with reference levels of baseline, female, and PBO + ESC respectively.

To test hypothesis 4, we utilized multiple linear regression to model HAMD-17 total score at week 8 (dependent variable), according to baseline SIRI (main independent variable). This model was adjusted for sex, age, BMI, treatment arm, and HAMD-17 at baseline. The nominal variables sex and treatment arm were dummy coded variables with male and ESC + PBO as reference levels, respectively. We screened the model for interactions between these covariates and baseline SIRI. The model was then fitted using backwards selection. A retrospective power analysis was conducted using parameters of valid N = 43, α = 0.05, and power = 0.80 which yielded R = 0.4124439 and R^2^ = 0.170. 

## 3. Results

### 3.1. Sample Characteristics and Group Comparisons (See Table 1)

Our sample with available CBC data consisted of 52 TRBDD subjects and 32 HCs. Amongst the TRBDD subjects, there were N = 23 in the placebo (ESC + PBO) arm and N = 29 in the combination treatment arm (ESC + CBX). Keeping in mind that HCs only had demographic and CBC-related data, there were no group differences in sex, age, baseline HAMD-17 total score, CBC markers, inflammatory cytokines, growth factors, and kynurenines that maintained significance after Benjamini–Hochberg correction. However, post-hoc pairwise comparisons revealed a lower BMI in HCs compared with TRBDD subjects in either treatment arm (*p* = 0.015). TRBDD subjects in the CBX + ESC arm trended lower HAMD-17 at week 8 compared to the PBO + ESC arm (*p* = 0.015). 

**Table 1 jpm-13-01408-t001:** Comparison of patient demographic, clinical, and biomarker variables by clinical subgroup.

Variable	N	Overall, N = 84 ^1^	HC, N = 32 ^1^	TRBDD (ESC + PBO), N = 23 ^1^	TRBDD (ESC + CBX), N = 29 ^1^	*p*-Value ^2^	q-Value ^3^
Sex	83					0.016	0.13
Male		44 (53%)	11 (34%)	16 (73%)	17 (59%)		
Female		39 (47%)	21 (66%)	6 (27%)	12 (41%)		
Age	74	40 (31, 52)	37 (26, 53)	47 (35, 58)	38 (31, 44)	0.083	0.42
BMI	71	3.37 (3.23, 3.49)	3.20 ^a^ (3.12, 3.34)	3.44 ^b^ (3.33, 3.60)	3.39 ^b^ (3.29, 3.57)	<0.001	0.017
Depressive severity							
HAMD-17 (baseline)	45	24.0 (20.0, 29.0)	-	23.5 (21.0, 26.0)	24.0 (20.0, 30.0)	0.66	0.87
HAMD-17 (Week 8)	45	10 (7, 17)	-	12 (9, 18)	8 (5, 13)	0.007	0.092
CBC-related markers							
Neutrophils	81	1.25 (1.03, 1.57)	1.15 (1.02, 1.53)	1.50 (1.06, 1.68)	1.28 (1.08, 1.49)	0.3	0.68
Monocytes	82	−0.92 (−0.92, −0.51)	−0.92 (−0.92, −0.65)	−0.69 (−1.13, −0.40)	−0.69 (−0.92, −0.51)	0.81	0.92
Lymphocytes	82	0.59 (0.47, 0.79)	0.59 (0.52, 0.83)	0.59 (0.47, 0.79)	0.59 (0.52, 0.79)	0.9	0.98
SIRI	81	−0.16 (−0.51, 0.24)	−0.28 (−0.54, 0.19)	0.04 (−0.32, 0.40)	−0.03 (−0.52, 0.20)	0.41	0.73
Pro-inflammatory cytokines							
IL-1α	21	0.69 (0.69, 0.79)	-	0.74 (0.69, 0.80)	0.69 (0.69, 0.69)	0.23	0.68
IL-1β	22	0.69 (0.69, 0.88)	-	0.69 (0.69, 0.96)	0.69 (0.69, 0.69)	0.43	0.73
IL-6	21	1.25 (1.00, 1.44)	-	1.56 (1.01, 1.70)	1.22 (1.00, 1.34)	0.18	0.64
TNF-α	21	1.18 (1.11, 1.40)	-	1.21 (1.12, 1.41)	1.18 (1.08, 1.30)	0.66	0.87
IFN-γ	21	0.69 (0.69, 0.69)	-	0.69 (0.69, 0.82)	0.69 (0.69, 0.69)	>0.99	>0.99
CRP	15	1.61 (1.16, 1.95)	-	1.58 (1.22, 1.65)	1.95 (1.19, 2.58)	0.3	0.68
MCP1	21	4.59 (4.48, 4.85)	-	4.76 (4.47, 4.90)	4.56 (4.48, 4.71)	0.47	0.73
Growth factors							
FGF	15	1.18 (0.88, 1.58)	-	1.36 (1.08, 1.57)	1.11 (0.76, 1.58)	0.73	0.87
VEGF	31	3.49 (3.36, 3.81)	-	3.41 (3.30, 3.49)	3.63 (3.48, 3.92)	0.057	0.36
EGF	21	1.28 (0.69, 1.71)	-	1.43 (1.13, 1.61)	1.20 (0.69, 1.71)	0.46	0.73
Kynurenines							
TRP	41	9.70 (9.51, 9.79)	-	9.59 (9.45, 9.76)	9.73 (9.58, 9.80)	0.16	0.64
KYN	41	5.76 (5.56, 5.94)	-	5.74 (5.64, 6.07)	5.76 (5.55, 5.86)	0.71	0.87
QUIN	41	4.02 (3.76, 4.29)	-	4.04 (3.83, 4.37)	3.98 (3.64, 4.25)	0.39	0.73
PIC	41	3.17 (2.84, 3.33)	-	3.20 (2.69, 3.36)	3.00 (2.84, 3.33)	0.97	>0.99
KYN/TRP	41	0.703 (0.701, 0.706)	-	0.705 (0.702, 0.707)	0.702 (0.701, 0.706)	0.25	0.68
QUIN/PIC	41	1.56 (1.41, 1.76)	-	1.55 (1.47, 1.71)	1.56 (1.39, 1.76)	0.63	0.87

^1^ Median (IQR) or Frequency (%). ^2^ Pearson’s Chi-squared test; Kruskal–Wallis rank sum test. ^3^ Benjamini–Hochberg correction for multiple testing. Note: superscript letters indicate post-hoc pairwise comparisons.

### 3.2. Univariate Relationships of Sample Characteristics with Baseline SIRI (See Table 2)

As part of the exploratory analysis, we assessed univariate Spearman’s correlations between CBC-based markers at baseline (neutrophils, monocytes, lymphocytes, and SIRI) and demographic, clinical, and biomarker characteristics. The relationships with baseline SIRI following the Benjamini–Hochberg (BH) adjustment included a negative correlation with MCP1 (rho = −0.609, *p* = 0.022) and a trend with age (rho = 0.319, *p* = 0.065). The pre-adjusted tests revealed trending correlations between baseline SIRI and baseline HAMD-17 (rho = 0.082, *p* = 0.055), lower IL-1α (rho = −0.460, *p* = 0.059), and KYN (rho = 0.691, *p* = 0.098). There were several other correlations worth noting for their significance prior to BH correction, as follows. Baseline neutrophils correlated with KYN (rho = 0.838, *p* = 0.022). Baseline monocytes correlated with KYN (rho = 0.522, *p* = 0.041). Finally baseline lymphocytes correlated with IL-6 (rho = 0.586, *p* = 0.021), EGF (rho = 0.628, *p* = 0.046), and KYN (rho = 0.261, *p* = 0.039). 

**Table 2 jpm-13-01408-t002:** Spearman’s correlations of CBC-related variables with demographics, HAMD-17 (baseline and week 8), and baseline biomarkers.

	Spearman’s Correlations of CBC-Related Markers (Baseline) with Demographics, HAMD17 (Baseline and Week 8), and Biomarkers (Baseline)											
	Spearman’s Coefficient (Rho)				*p*-Value (Unadjusted)				*p*-Value (Benjamini–Hochberg)			
	Neutrophils	Monocytes	Lymphocytes	SIRI	Neutrophils	Monocytes	Lymphocytes	SIRI	Neutrophils	Monocytes	Lymphocytes	SIRI
Age	0.470	0.442	−0.011	0.319	0.172	0.575	0.392	0.325	0.732	0.727	0.975	0.065
BMI	0.387	0.712	0.151	0.436	0.783	0.442	0.184	0.227	0.922	0.966	0.550	0.548
HAMD17 (Baseline)	0.082	−0.435	−0.225	−0.114	0.296	0.106	0.921	0.055	0.505	0.915	0.694	0.496
HAMD17 (Week 8)	0.444	0.312	−0.076	0.338	0.815	0.292	0.497	0.676	0.966	0.708	0.837	0.922
IL-1α	−0.460	−0.526	−0.399	−0.396	0.153	0.172	0.680	0.059	0.505	0.505	0.922	0.343
IL-1β	−0.558	−0.227	0.193	−0.557	0.307	0.664	0.596	0.138	0.708	0.922	0.908	0.503
IL-6	0.087	0.323	0.586	−0.173	0.628	0.384	0.021	0.225	0.915	0.764	0.171	0.604
TNF-α	0.178	0.297	0.456	−0.032	0.791	0.130	0.053	0.637	0.966	0.496	0.322	0.916
IFN-γ	−0.203	−0.084	−0.075	−0.202	0.911	0.217	0.070	0.675	0.975	0.595	0.372	0.922
CRP	0.173	0.247	0.128	0.127	0.172	0.196	0.529	0.503	0.505	0.550	0.857	0.837
MCP1	−0.305	−0.465	0.307	−0.609	0.060	0.099	0.339	0.001	0.344	0.447	0.727	0.022
FGF	0.560	0.659	0.540	0.464	0.949	0.691	0.658	0.760	0.975	0.922	0.922	0.962
VEGF	−0.387	−0.527	0.014	−0.455	0.313	0.692	0.178	0.158	0.716	0.922	0.517	0.505
EGF	−0.244	0.014	0.628	−0.440	0.862	0.352	0.046	0.950	0.975	0.732	0.289	0.975
TRP	0.050	−0.327	−0.403	0.200	0.688	0.874	0.711	0.844	0.922	0.975	0.926	0.975
KYN	0.838	0.522	0.261	0.691	0.022	0.041	0.039	0.098	0.180	0.278	0.278	0.447
QUIN	0.310	0.261	−0.137	0.327	0.337	0.113	0.614	0.170	0.727	0.474	0.915	0.505
PIC	−0.694	−0.245	0.167	−0.579	0.158	0.834	0.599	0.335	0.505	0.975	0.908	0.727
KYN/TRP	0.519	0.598	0.421	0.282	0.164	0.148	**0.076**	0.304	0.505	0.505	0.391	0.708
QUIN/PIC	0.642	0.323	0.128	0.518	0.626	0.218	0.290	0.843	0.915	0.595	0.708	0.975

All variables except age have been log transformed to meet assumptions of normality.

### 3.3. SIRI—to—HAMD17 Relationships between Timepoints (See Table 3)

There was no significant mean difference in SIRI between baseline and week 8. On a robust linear mixed-effects model adjusted for demographics, HAMD-17 was significantly associated with SIRI at week 8 compared to baseline (β = 4.73, 95% CI [0.73–8.74], *p* = 0.02). 

**Table 3 jpm-13-01408-t003:** Robust linear mixed-effects model of HAMD17 according to timepoint.

Outcome Variable: HAMD 17 Total			
*Predictors*	*Estimates*	*CI*	*p*
(Intercept)	17.55	9.97–25.13	**<0.001**
Sex (Female)	−0.11	−2.68–2.47	0.936
Age	0.1	−0.00–0.20	0.057
BMI	0	−0.23–0.23	0.991
Treatment arm (ESC + CBX)	1.33	−1.36–4.03	0.331
SIRI (log)	−2.04	−4.95–0.86	0.168
Timepoint (Week 8)	−12.33	−14.77–−9.88	**<0.001**
SIRI (log) * Timepoint	4.73	0.73–8.74	**0.02**
(Week 8)			
**Random Effects**			
σ^2^	30.17		
τ_00 Timepoint_	0		
ICC	0		
N _Timepoint_	2		
Observations	84		
Marginal R^2^/Conditional R^2^	0.605/0.605		

BMI and SII (baseline) are log transformed to meet assumption of normality. Sex, treatment arm, and timepoint are dummy-coded variables with “male”, “ESC + PBO”, and “baseline” as reference levels, respectively. * Indicates interaction between two variables.

### 3.4. Post-Treatment HAMD-17 by Pre-Treatment SIRI

On multiple regression modelling of HAMD-17 (week 8), there were significant associations with female sex (*p* = 0.043) and ESC + CBX treatment arm (*p* = 0.025), but there was no independent effect of SIRI (baseline) (*p* = 0.196) (Table 4). However, there was a significant interaction effect of SIRI (baseline) and HAMD-17 (baseline) (β = 0.91, 95% CI [0.24–1.59], *p* = 0.009) in the final model adjusted for interactions (R^2^/R^2^ adjusted = 0.467/0.361) (Table 5).

## 4. Discussion

There is mounting evidence for immune–metabolic dysregulation in depressive disorders, as well as growing interest in CBC-based markers as indices of systemic inflammatory response across medical fields including psychiatry. Based on the paucity of studies (to date) of CBC-based inflammatory indices in bipolar and TRBDD specifically, we conducted a secondary biomarker analysis on a prior RCT by Halaris et al. (2020) which established support for adjunctive CBX in TRBDD [18]. Towards that end, we tested several exploratory (not confirmatory) hypotheses based on available knowledge of immune–metabolic dysregulation in unipolar and bipolar depression, as well as emerging reports of CBC-related indices in the context of bipolar disorder. 

We were unable to support hypothesis 1, that baseline SIRI can discriminate group differences between TRBDD and HCs. This was somewhat unexpected, in part because we previously established that the classical pro-inflammatory marker CRP was elevated in TRBDD compared to HCs [19]. However, in hypothesis 2 we also applied correlational analysis to preliminarily infer biological associations between baseline SIRI and classical pro-inflammatory markers (IL-1α, IL-1β, CRP, TNF-α, IFN-γ, and IL-6) but none were detected in our sample. Granted, there was an inverse relationship between baseline SIRI and MCP-1, which is a potent chemokine that attracts circulating monocytes to the site of inflammation [53]. The general lack of association between SIRI and pro-inflammatory cytokines is nonetheless informative, and may partly explain why SIRI did not discriminate TRBDD from HCs. While it does not discount the presumptive pro-inflammatory significance of SIRI, the absence of correlations with canonical inflammatory cytokines raises questions about the nuances of CBC-based indices. For example, SIRI may capture more dynamic “state” based differences within the TRBDD sample (rather than “trait” or group differences). In fact, results from hypothesis 3 may be supportive here since, amongst TRBDD subjects, the difference in the SIRI—to—HAMD17 relationship changed significantly within patients across treatment timepoints. While this does not necessarily represent a treatment effect, per se, the contrasting significance of the SIRI—to—HAMD17 relationship over the treatment course supports the notion of the dynamic “state”-based significance of SIRI in the context of the clinical course of TRBDD. 

For hypothesis 4 we explored the relationship of pre-treatment SIRI with post-treatment clinical outcomes, to assess a potential “moderating” role of baseline SIRI. We justified this preliminary hypothesis based on broader literature support that immune–metabolic dysregulation is a component of depressive neuroprogression, including in treatment-refractory forms of depressive illness. Prior studies in our TRBDD sample also revealed that treatment response to CBX augmentation were associated with pre-treatment abnormalities in inflammatory markers [22,23]. There were no *independent* effects of baseline SIRI on HAMD-17 (week 8) on either univariate or multivariate levels. Conceptually, the lack of an independent effect of baseline SIRI on depressive outcomes was not unexpected, for several reasons, including the fact that SIRI was not clearly associated with classical pro-inflammatory markers in our sample to begin with and the plausible context-dependent significance of SIRI itself. We therefore incorporated a focused interaction screen on baseline SIRI with pre-existing covariates that were rationally selected, even though they were insignificant in the non-interaction model. What we found was that the relationship between pre- and post-treatment depressive severity, which only trended significance in the non-interaction model, became strongly significant when considering baseline SIRI. Expressed differently, the effect of baseline depressive severity on treatment outcomes was dependent on baseline SIRI (Figure 1). Amongst subjects with higher baseline HAMD-17 (>18 total score), those with elevated baseline SIRI had higher HAMD-17 at week 8 and vice versa. While it may be premature to infer translational significance outside of this clinical context, the interaction between pre-treatment depression and SIRI supports the general concept that biological markers can supplement and enhance our understanding of clinical predictors, particularly in the psychiatric context. 

Why is it plausible that baseline SIRI might moderate the predictive relationship between pre- and post-treatment depressive severity? This question fits the pre-existing conceptual framework for thinking about treatment refractoriness in a neuroprogressive context. SIRI-related mechanisms may contribute to our understanding of the underlying biological “vulnerabilities” which already appear to involve endocrine, neurovascular, and immune domains [8,54,55,56]. The conceptual significance of SIRI here may fit the ongoing discussion on the “cross-talk” between peripheral and CNS inflammation as it relates to depressive etiology [57]. Dysfunction at the vascular–endothelial interface is increasingly relevant here, especially considering reports of abnormal vascular endothelial growth factor (VEGF) in treatment-refractory depression [58,59,60,61,62,63,64]. SIRI adds to this discussion by bringing to bear the aspect of activated circulating immune cells (specifically monocytes, which are factored into SIRI), which are known to undergo trans-endothelial migration to promote microglial activation and thus neuroinflammation [28,65,66,67,68,69]. While we found some preliminary trends between SIRI (and individual cell counts therein) and a focused set of immune–metabolic markers, clearly further studies are needed to establish, without the constraints of a secondary retrospective study, the clinical and biological significance of SIRI in TRBDD and bipolar depression. 

## 5. Conclusions and Limitations

We presented a retrospective biomarker analysis for the purposes of testing a preliminary hypothesis (exploratory, not confirmatory) for the biological and clinical relevance of SIRI in the TRBDD context. Thus, we constructed our preliminary hypotheses to balance exploratory goals and the limitations of post-hoc study design. To support interpretive rigor, we provided retrospective power analyses and FDR adjustments, so that the reader can interpret pre-and post-adjusted trends in this exploratory context. From an analytic standpoint, it is worth noting that there was no significant interaction between the main intervention in the clinical study (ESC + CBX) and baseline SIRI on treatment outcomes. Further, we lacked power to test whether the baseline SIRI—to—HAMD-17 interaction was dependent on the treatment arm and could therefore be interpreted as a general effect of either treatment arm (note that ESC was present in both arms). Methodologically, we also acknowledge that TRBDD patients were not subdivided by bipolar disorder type 1 versus type 2. While this distinction is certainly more relevant for the (hypo)manic or mixed phases of bipolar illness, which was not the objective of Halaris et al. (2020) which focused on bipolar depression [18], we cannot discount the possible biological relevance of this clinical distinction even for the depressed phase in TRBDD. An additional consideration is that there was no independent adjustment for number of prior depressive episodes or comorbid anxiety; however, we note that the TRBDD designation was assigned based on the multidimensional assessment with the MSM which accounts for duration of the current depressive episode. Future studies might also take into account emerging consensus criteria for TRBDD that include broader criteria such as failure of psychotherapy and/or electroconvulsive therapy [70]. Taken together, this study sheds light on the putative role of elevated SIRI as a useful predictor of poor treatment response amongst TBRDD subjects with elevated baseline depression. Translational psychiatry will continue to benefit from combined clinical and biological factors, including the area of cell-based peripheral markers of inflammation, which evidently play a role in both phases of bipolar illness, including TRBDD.

## Figures and Tables

**Figure 1 jpm-13-01408-f001:**
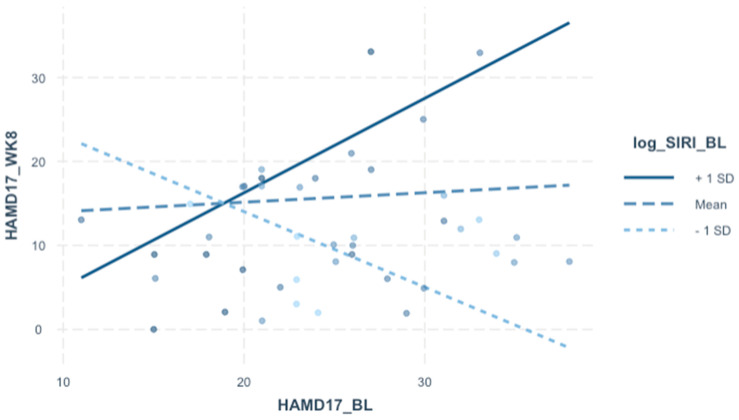
**HAMD-17 (Week 8) associated with SIRI (baseline)—to—HAMD-17 (baseline) interaction.** Interaction plot depicting the effect of HAMD-17 (baseline) × log SIRI (baseline) interaction on HAMD-17 (Week 8), from Table 5.

**Table 4 jpm-13-01408-t004:** Multiple regression model of HAMD17 (week 8) according to SIRI (baseline) and relevant covariates, without interactions.

Outcome Variable: HAMD 17 Total (Week 8)			
*Predictors*	*Estimates*	*CI*	*p*
(Intercept)	−15.64	−56.10–24.83	0.438
Sex (Female)	4.39	0.15–8.63	**0.043**
Age	0.12	−0.06–0.29	0.198
log BMI	4.61	−6.91–16.14	0.422
Treatment arm (CBX + ESC)	−5.3	−9.89–−0.71	**0.025**
HAMD17 (baseline)	0.34	−0.02–0.69	0.061
log SIRI (baseline)	2.48	−1.34–6.29	0.196
Observations	43		
R^2^/R^2^ adjusted	0.352/0.244		

BMI and SIRI (baseline) are log transformed to meet assumption of normality. Sex and treatment arm is dummy-coded variables with ‘male’ and ‘ESC + PBO’ as reference levels, respectively.

**Table 5 jpm-13-01408-t005:** Multiple regression model of HAMD17 (week 8) according to SIRI (baseline) and relevant covariates, with interaction by baseline depression.

Outcome Variable: HAMD 17 Total (Week 8)			
*Predictors*	*Estimates*	*CI*	*p*
(Intercept)	−10.29	−47.75–27.17	0.581
Sex [Female]	6.02	1.93–10.11	0.005
Age	0.05	−0.12–0.22	0.525
log BMI	3.31	−7.35–13.96	0.533
Arm [CBX + ESC]	−5.13	−9.36–−0.90	0.019
HAMD17 (baseline)	0.42	0.09–0.75	0.014
log SIRI (baseline)	−17.85	−33.26–−2.44	0.024
HAMD17 (baseline) × log SIRI (baseline)	0.91	0.24–1.59	0.009
Observations	43		
R^2^/R^2^ adjusted	0.467/0.361		

BMI and SIRI (baseline) are log transformed to meet assumption of normality. Sex and treatment arm are dummy-coded variables with “male” and “ESC + PBO” as reference levels, respectively.

## Data Availability

We opted not to share our data.

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
