# Peer review of "Systemic Inflammatory Response Index (SIRI) at Baseline Predicts Clinical Response for a Subset of Treatment-Resistant Bipolar Depressed Patients"

_jpm, 2023, doi:10.3390/jpm13091408_

Round 1

Reviewer 1 Report

Authors have conducted a secondary analysis of the systemic inflammatory response index (SIRI = absolute neutrophils x absolute monocytes/absolute lymphocytes) in relation to other biomarkers and clinical outcomes after escitalopram combined with the COX-2 inhibitor, celecoxib, versus placebo. 

Introduction is well written and describe the prior RCT and need for this secondary analysis. Methods section is detailed and clear to understand and follow, they have clearly describe that additional data from healthy controls were used for this secondary analysis. Prior RCT has been described which is helpful for th readers to understand the premise of this study. Results  are presented in detail yet understandable fashion, clear associations and relationships with their caveats have been discussed. Similarly, in discussion section they have made appropriate comments on the findings, especially stating the dispcrepencaies in SIRI and other pro inflammatory biomarkers at baseline. Limitation are also appropriately discussed. One suggestion I would like to make is to change the title to give a better glance of actual findings, for example authors state that SIRI is only significant if the HAMD scores were higher in the TRBD population, this is important, as from the available data we cannot generalize the predictive nature of the SIRI in all TRBD patients.

Author Response

Dear Reviewer 1,

Thank you for your thoughtful feedback. I agree with your suggestion to tailor the title, so as not to lend to over-generalization. I will change it to something like this:

"Systemic inflammatory response index (SIRI) at baseline predicts clinical response for a subset of treatment-resistant bipolar depressed patients"

Thanks again for taking the time,

Stephen Murata, MD

Reviewer 2 Report

The current study investigated the association between the SIRI, inflammatory cytokines, and metabolites in the kynurenine pathway in TRBD patients. The possible predictive role of SIRI in TRBD treatment response was also explored. The authors found that baseline SIRI was negatively associated with post-treatment HAMD-17 scores and baseline SIRI and baseline HAMD-17 scores interaction were associated with post-treatment HAMD-17 scores. Although the study topic is interesting and important, there were some points need to be addressed.

1.     The major limitation of the current study was the modest sample size. Under the limited sample size, limited power was suspected. In addition, there were many different comparisons in the analysis, so the adjustment for multiple testing should be added.

2.     There were also many dependent variables in the regression model with a limited sample size, the stability of the statistic models should be considered.

3.     There were fewer cases that had baseline inflammatory markers, growth factor, and kynurenine pathway metabolites. The authors should explain the reasons.  

4.     The current study did not control for the effects of metabolic profiles and different mood stabilizers and antipsychotics on biomarkers presentation.

5.     What is the difference between the systemic inflammatory response index (SIRI) and the systemic inflammation index (SII) for the biological meaning?

6.     In Table 2, baseline HAMD-17 was significantly associated with lower IL-2 at baseline (p=0.035), higher TNF-α at week 8 (p=0.043), and lower quinaldehyde (QUINA) at week 8 (p=0.015). The results were confusing. Which are the dependent variables (DVs)? It is supposed that baseline HAMD-17 should be the DV, then using week 8 data as independent variables (IVs) would be inappropriate. (Page 9, line 259-260)

7.     The same questions in Table were also found. Which is the DV?

8.     Because of the limited sample size, putting too many variables and interaction terms in the multiple regression model may generate overfitting results, such as those in Table 8.

9.     In Table 8, there is no significant interaction effect of HAMD 17 baseline*log SIRI baseline*treatment arm. How to interpret the results?

10.  There was an error in the format in Page 4, line 181.

Author Response

Dear Reviewer 2, I'm attaching responses to your great feedback here. The adjustments to the analysis were significant and in line with yours and other reviewers suggestions, and the final implementation will be shown in the manuscript to be uploaded soon. 

Reviewer 3 Report

The analysis carried out by the authors is very detailed and reliable, and the research topic itself is justified from both a scientific and clinical point of view. I have the following comments.

1.       The introduction of the work is written too generally, an additional one or two paragraphs are needed to summarize in more detail the results of research on the markers of the inflammatory reaction in bipolar disorder -  research on pro-inflammatory cytokines, micronutrients (zinc, copper) or markers of oxidative stress (e.g. TBARS) (for example: DOI: 10.12740/PP/OnlineFirst/65250; DOI: 10.1016/j.jad .2015.10.026; DOI: 10.1111/acps.13508; DOI: 10.1016/j.bbih.2021.100232; DOI: 10.1007/s12035-016-0124-8)

2.       I don't see any information anywhere on what percentage of patients were people with type II and what percentage of patients with type I of bipolar disorder. Many data indicate biological differences of these subtypes and it would be worth including this variable in the analysis, where it is statistically possible. If the authors do not have such data, it is a serious limitation that should be listed in limitations.

3.       Due to the fact that in the study of biological markers of bipolar disorder, their level may be significantly affected by the biological advancement of the disease - often related to its duration - an additional variable, i.e. age at the onset and/or duration of the disease and/or the number of previous episodes of the disease, should be included in the statistical analyses. If the authors have any of these data, it will be a valuable supplement to the analysis. If they do not, this is a significant limitation of the study and a significant weakening of the reliability of the analyzes, which should be listed under limitations

4.       In the information on ethical issues and obtaining informed consent, the number of the bioethics committee approval is missing.

Author Response

Dear Reviewer 3, 

I really appreciated your feedback. Here are my specific responses:

  1. I tightened up the introduction, to include a pre-amble on the concept of neuroprogression in bipolar disorder, and its putative role in treatment-refractory forms of depression generally. In that context I incorporated a brief discussion of the emerging neurobiological correlates of bipolar neuroprogression (inflammatory, metabolic) including the citations you provided. The introduction was also enhanced to better set up the primary hypothesis, that baseline immune dysregulation moderates treatment response in TRBDD. Thank you.
  2. Good point--I agree, the biological endophenotypes of bipolar I and II may be distinct. We will include this in the limitations. We did not have a structured interview to screen for formal diagnoses of type 1 vs 2, nor available retrospective data on their prior admission diagnoses that will enable us to infer bipolar 1 vs. 2. This will therefore be included in the limitation section as suggested. 
  3. Another good point. We do not have data related to duration of un(der)treated bipolar depression, including date of initial bipolar diagnosis, number of prior bipolar depressive episodes. This is important but challenging, given the known delay of bipolar diagnosis relative to initial unipolar depression diagnosis (base rate average 8 years delay in diagnosis in many studies). Comorbidities is another factor. This will be touched on in the limitations as well.
  4. We will include a statement on our ethical protocol, which was missing. Thank you for the catch.

Sincerely,

Stephen Murata, MD, PGY4

Reviewer 4 Report

In addition to highlighting the significance of using cell-based peripheral indicators of inflammation in the research of bipolar disease, the paper discusses the potential function of SIRI as a predictive biomarker for poor treatment response in TRBDD patients. The findings offer information on putative molecular vulnerabilities underlying TRBDD and add to the expanding body of literature correlating immune dysregulation with treatment resistance in depression. Several minor modifications should be made to the manuscript.

 1.      It is challenging to determine the precise features of the study population due to the unclear criteria for participant selection. Participants with BD I and BD II were either included, or only one of the two, however this is not stated explicitly. It should be stated as these two have different characteristics.

2. The method section refers to a primary study for full details of the study design but does not provide a reference or citation to access that study in every point mentioned. It is essential to include proper references for readers to access the original study in every point mentioned.

3. The method section neither explains nor provides a power analysis to support the sample size of 69 individuals. To ensure that the study is sufficiently powered to detect meaningful effects, clear sample size justification is required.

4. 4.   The paper's discussion appears to be devoid of a critical examination of the study's limitations. While the section "Conclusions & limitations" briefly emphasizes the need for a larger sample size, it would be beneficial to focus on other potential limitations, such as the study design, potential biases, and generalizability of the findings.It is important to acknowledge that the study may have confounding variables or uncontrolled factors that could impact the results. Discussing these potential confounders and how they were accounted for (if at all) would add credibility to the study's conclusions.

5. While the study hints to potential translational psychiatric consequences, it should be more explicit in addressing how the findings contribute to clinical practice or future research paths.

6.   More direct comparisons with pertinent current literature might enhance the discussion. Discussing how the study's findings fit with or differ from past studies would help readers contextualize the findings.

Author Response

Dear Reviewer 4,

Thank you for taking the time to provide this insightful feedback. Here are my responses, and how I assimilated your input:

  1. Thank you for this question. In the original archives we have data for bipolar I vs. II, but not in the present dataset for Halaris (2020) which this secondary study is based on. This clinical designation was not included due to the primary focus on the depressed phase.  For memory, the proportion was approximately 65% BP1 vs. 35% BP2. From what the literature indicates, yes, there are different biosignatures when comparing depressed vs. (hypo)manic/mixed phases of bipolar, but it would be interesting to explore whether biological endophenotypes differ in the depressed phases of bipolar I vs. II. Taken together, this limitation will be included this in the limitations section. 
  2. We will include references to Halaris et. al (2020) at every point of reference, thank you for the catch. 
  3. In the updated draft, we include a retrospective power analysis for our modeling steps. Our power analysis is based on n=43 (valid N for TRBDD subjects with available SIRI, demographic, and clinical data), significance level = 0.05, and power threshold = 0.8. This calculation yields R=0.412, from which we calculate R^2 = 0.412^2 = 0.17. The adjusted R^2 for our final models (0.244 and 0.361) both which exceed the expected 0.17 figure above. As a sanity check, based on the lower R2=0.244 from the non-interaction model, sig level = 0.05 and n=43, the calculated power is 0.93. We also included some extra commentary about the stability/fit of those models.
  4. The study limitation section was weak. It has been updated to be much more specific and conservative. Of note, in my revision I refined the analysis to be more disciplined to the key hypothesis, which allowed us to identify extraneous testing steps. I also included a post-hoc bonferroni analysis at the univariate testing level, to minimize the false detection rate. These adjustments indirectly implicate the limitation section, where the specific points will be discussed specifically.
  5. Yes, we do hint at translational value. On one hand, I added language to make this more conservative, that is to say, the results are not ready for clinical "prime time". Rather, given the limitations of a post-hoc secondary analysis, we have uncovered the potential biological relevance of SIRI in a clinical context, which will require much more validation in future studies. 
  6. In the discussion section, I have included more direct comparisons to literature. To be specific, I focused on studies in support of a "moderating" role of baseline CBC-based indices in relation to a post-treatment outcome. I distinguished between studies describing changes in biomarker over time, that may have a more direct "mediating" role in treatment. 

The finalized edited manuscript will follow shortly. 

Thank you again,

Stephen Murata, MD, PGY4

Reviewer 5 Report

1) The operational definition of TRBD adopted by the authors is controversial from a clinical standpoint. Moreover, they just mention the Maudsley staging method (MSM) without providing punctual references for that. Specifically, they failed to mention either: https://pubmed.ncbi.nlm.nih.gov/19457299/   or https://pubmed.ncbi.nlm.nih.gov/19192471/ .  The most problematic issue here is that, albeit relevant (as it introduced a multi-dimensional approach to TRD), the MSM does not fit the complexity of TRBD. Instead, two alternative references should have been regarded by the authors, and should now appear in the edited version of the critical discussion: https://www.sciencedirect.com/science/article/abs/pii/S0165032720325544   and https://pubmed.ncbi.nlm.nih.gov/30520709/. Specifically, operational definitions poorly sound from a clinical perspective (e.g., TRBD upon failure of two different trials with antidepressant monotherapy) hamper the rationale of the whole study and the significance of the findings.

2) The sample size is not sufficient to allow for the analysis to be carried out, especially for multivariate analysis. I found no power and sample size a-priori calculation.

Author Response

Dear Reviewer 5:

We have incorporated changes based on both of your feedback items:

  1. We were missing Fekuda et. al citations (omission) which were both added. We also incorporated the citations you suggested regarding TRBD definition, which I agree is important based on the clinical complexity at play. I also added a helpful detail from the primary study, "In our study, 80% of the patients scored between 5 and 8 [on the MSM], indicating moderate to severe treatment-resistance."

  2. Yes! I agree with the reviewer that limited sample size prompts concern for power. We will adjust the manuscript to include the retrospective power analysis based on N=43 (the subset of TRBDD patients with available observations):

      approximate correlation power calculation (arctangh transformation)

    • n = 43
    • r = 0.4124439
    • level = 0.05
    • power = 0.8
    • alternative = two.sided

    Based on R=0.4124439, we can calculate R2 = 0.4124439 ^2 = 0.17011. So based on N=43, we would expect the a given independent variable to explain ~17% variance in the outcome (HAMD17 week 8) with the above parameters. In the model unadjusted for interaction (table 7***), the adjusted R2 = 0.244 suggesting SIRI (baseline) accounts for 24.4% of the variance in HAMD17 (week 8) which is greater than the 17% threshold calculated above. In the model adjusted for interaction (table 8), adjusted R2 = 0.361, suggesting that the SIRI*HAMD17 (baseline) interaction term accounts for 36.1% of the variance in HAMD17 (week 8) which is also above the 17% threshold. For a ‘sanity check’ - if we take the smaller of these two effect sizes (R2=0.244, where R=0.493) to make a conservative power estimate, that yields ~93.3% power:

    approximate correlation power calculation (arctangh transformation)

    • n = 43
    • r = sqrt(0.244)
    • level = 0.05
    • power = 0.9318081
    • alternative = two.sided

    So, taken together N=43 seems to be adequate sample size for the retrospective power analysis, keeping in mind limitations of post-hoc power analysis generally. The statistical analysis section has been updated to reflect the pertinent details of this retrospective power analysis.

Thank you for taking the time to help us on this manuscript. The revised version will be uploaded by the end of the day. 

Sincerely,

Stephen Murata, MD, PGY4

Round 2

Reviewer 2 Report

The authors revised the manuscript thoroughly, and I have no more suggestions. 

Reviewer 3 Report

Authors made satisfactory corrections due to reivewers comments and now article is suitable for publication 

Reviewer 5 Report

Thank you for your diligent revision